# Comprehensive clinical assessment identifies specific neurocognitive deficits in working-age patients with long-COVID

David A. Holdsworth[1,2,3]*, Rebecca Chamley[2,4], Rob Barker-Davies[5,6], Oliver O'Sullivan[5], Peter Ladlow[5,7], James L. Mitchell[5,8], Dominic Dewson[5], Daniel Mills[5], Samantha L. J. May[5], Mark Cranley[9], Cheng Xie[3], Edward Sellon[3,9], Joseph Mulae[1], Jon Naylor[1], Betty Raman[4], Nick P. Talbot[3,10], Oliver J. Rider[4], Alexander N. Bennett[5,11], Edward D. Nicol[1,12,13]

1 Royal Centre for Defence Medicine Birmingham, Birmingham, United Kingdom, 2 Academic Department of Military Medicine, Birmingham, United Kingdom, 3 Oxford University Hospitals NHS Foundation Trust, Oxford, United Kingdom, 4 University of Oxford, OCMR, Division of Cardiovascular Medicine, Oxford, United Kingdom, 5 Defence Medical Rehabilitation Centre, Academic Department of Military Rehabilitation, Stanford Hall, United Kingdom, 6 Loughborough University, School of Sport, Exercise and Health Sciences, Loughborough, United Kingdom, 7 Department for Health, University of Bath, Bath, United Kingdom, 8 University of Birmingham, Metabolic Neurology, Institute of Metabolism and Systems Research, Birmingham, United Kingdom, 9 Defence Medical Rehabilitation Centre, Stanford Hall, United Kingdom, 10 Department of Physiology, University of Oxford, Anatomy and Genetics, Oxford, United Kingdom, 11 Imperial College London National Heart and Lung Institute, London, United Kingdom, 12 Department of Cardiology, Royal Brompton and Harefield NHS Foundation Trust, London, United Kingdom, 13 School of Biomedical Engineering and Imaging Sciences, Kings College, London, United Kingdom

* david.holdsworth@nhs.net

**Data Availability Statement:** Data cannot be shared publically due to the connection to UK

## Abstract

### Introduction

There have been more than 425 million COVID-19 infections worldwide. Post-COVID illness has become a common, disabling complication of this infection. Therefore, it presents a significant challenge to global public health and economic activity.

### Methods

Comprehensive clinical assessment (symptoms, WHO performance status, cognitive testing, CPET, lung function, high-resolution CT chest, CT pulmonary angiogram and cardiac MRI) of previously well, working-age adults in full-time employment was conducted to identify physical and neurocognitive deficits in those with severe or prolonged COVID-19 illness.

### Results

205 consecutive patients, age 39 (IQR30.0–46.7) years, 84% male, were assessed 24 (IQR17.1–34.0) weeks after acute illness. 69% reported ≥3 ongoing symptoms. Shortness of breath (61%), fatigue (54%) and cognitive problems (47%) were the most frequent symptoms, 17% met criteria for anxiety and 24% depression. 67% remained below pre-COVID performance status at 24 weeks. One third of lung function tests were abnormal, (reduced lung volume and transfer factor, and obstructive spirometry). HRCT lung was clinically

Defence personnel. However, all reasonable requests for anonymised data will be considered and responded to positively. Brigadier Duncan Wilson (duncan.wilson651@mod.gov.uk) Medical Director to the Surgeon General, Defence Medical Services Caldecott Guardian has agreed to act as the non-author institutional contact for any approaches.

**Funding:** The authors received no specific funding for this work.

**Competing interests:** The authors have declared that no competing interests exist.

indicated in <50% of patients, with COVID-associated pathology found in 25% of these. In all but three HRCTs, changes were graded 'mild'. There was an extremely low incidence of pulmonary thromboembolic disease or significant cardiac pathology. A specific, focal cognitive deficit was identified in those with ongoing symptoms of fatigue, poor concentration, poor memory, low mood, and anxiety. This was notably more common in patients managed in the community during their acute illness.

## Conclusion

Despite low rates of residual cardiopulmonary pathology, in this cohort, with low rates of premorbid illness, there is a high burden of symptoms and failure to regain pre-COVID performance 6-months after acute illness. Cognitive assessment identified a specific deficit of the same magnitude as intoxication at the UK drink driving limit or the deterioration expected with 10 years ageing, which appears to contribute significantly to the symptomatology of long-COVID.

## Introduction

Post-COVID illness has become a common and disabling complication of infection with SARS-CoV-2. As such, the health and social impacts of post-COVID illnesses are likely to be substantial, and whilst the focus of management has initially revolved around acute illness, the longer-term effects are equally, if not more, important to pandemic recovery. Persistent symptoms 12 weeks after acute COVID-19 infection ranges from 2.3 to 10% [1,2]. In a UK sample (as at 06 Jan 2022), the Office for National Statistics reports that 890,000 UK citizens now have long COVID symptoms >12 weeks after confirmed or presumed COVID-19 illness [3]. Whilst incomplete data for laboratory-confirmed infection make the interpretation of these self-reported symptoms challenging, the potential impact of post-COVID illness is significant, with symptoms limiting work and family life, in a working-age population, many of whom have no pre-morbid health conditions. In line with this, the National Institute for Health Research (NIHR) second themed review surveyed 3,286 respondents with long-COVID, 80% of whom reported that their symptoms were interfering with their work and 85% had sought access to healthcare [4].

From the earliest reports, the symptoms described in post-COVID syndromes were non-specific, and included fatigue, breathlessness and cognitive impairment [5–7]. Studies emerged, which reported the neuropsychiatric and cognitive data more systematically. Mazza et al. reported the results of neuropsychiatric screening of an Italian cohort of 402 patients one month after hospital discharge with acute COVID-19. The proportions of patients with anxiety, depression and post-traumatic stress in the pathological range were 42%, 31% and 28% respectively [2]. 56% of the cohort registered in the pathological range for at least one neuropsychiatric domain. The same investigators have since reported the findings of patients at 6- and 12-month post COVID-19. The proportion with any neuropsychiatric score in the pathological range was 44% and 45% respectively [3]. Becker et al. reported the outcomes of cognitive function testing in a New York cohort of 740 patients (50% managed in the community setting). The most obvious cognitive defects were in memory encoding (24% affected); category fluency (20%); processing speed (18%) and executive function (16%) [8].

Cognitive and neuropsychiatric symptoms may result from direct viral injury to multiple organs, persistent activation of systemic or localised inflammatory pathways, or from the

understandable health anxiety and distress resulting from illness and external factors including the global pandemic. There is certainly now clear evidence that the COVID-19 pandemic has increased levels of anxiety and depression, regardless of physical illness and infection [9]. There is also emerging evidence in mouse models, human post-mortem studies and now (as pre-print) in humans (in vivo) of neuro-inflammation, cellular dysregulation, impaired neuro-genesis and demyelination, even following mild acute COVID-19 illness [10–12].

Early in the global COVID-19 pandemic, the UK Defence Medical Services (DMS) identi-fied a need to provide thorough, standardised clinical assessment of serving personnel (SP) who had recovered from severe (hospitalised) COVID-19 illness, or who were experiencing prolonged (>12 weeks) symptoms [13]. The service personnel in these groups were occupa-tionally restricted because of their illness and were unable to undertake their usual work. SP are a comparatively fit adult population in full time employment, with a low prevalence of background medical conditions. As part of their service, they must pass an annual physical fit-ness test, and they may be required to exercise at maximal intensity and to perform tasks of a safety-critical nature in potentially dangerous situations and/or remote environments. Conse-quently, there is a duty of care to both identify and manage significant organ pathology, or to provide robust assurance that no pathology exists, prior to a return to high-intensity physical exercise and/or high-risk environments.

This comprehensive clinical pathway for affected personnel included data on subjective symptoms, patient-reported outcome measures (PROMs) for anxiety, depression, stress, wellbe-ing, and fatigue, and a physician-supervised objective measure of cognitive performance in every patient. These measurements were collected alongside objective measurements of func-tional capacity and cardiopulmonary performance to identify/exclude clinically significant physical pathology, such as lung fibrosis, pulmonary embolus, or myocarditis with LV impairment, that could explain symptoms, and then to either direct appropriate clinical man-agement for any pathology identified, or to provide reassurance and direct attention towards rehabilitation and recovery. Cardiopulmonary exercise testing (CPET) in a ramp protocol to peak exercise was deliberately placed at the centre of clinical evaluation for abnormalities of pul-monary or cardiac function, as it is a sensitive detector of cardiopulmonary abnormalities, the gold-standard measurement of cardiorespiratory fitness, and is recommended in the diagnosis of unexplained exertional limitation [11–13]. Exercise capacity, no matter how measured, is a powerful prognostic predictor of all-cause mortality and has utility in assessment of individuals with occupational requirement for maximal exercise [14,15]. The objective was to identify phys-ical and neurocognitive deficits in those with severe or prolonged COVID-19 illness.

## Methods

### Patients

The Defence COVID-19 Recovery Service (DCRS) opened in August 2020. All service person-nel diagnosed with confirmed or probable COVID-19 were referred from primary care if they met criteria for severe acute COVID illness (defined as hospital admission); had prolonged symptoms after recovery from acute illness (>12 weeks); or had features associated with ele-vated clinical risk (chest pain and ECG changes, or elevated cardiac enzymes during acute ill-ness and desaturation at rest or on exertion in primary care). The criteria are presented in Table 1. Eligible patients were given the opportunity to volunteer for a military study of post-COVID disease (M-COVID). All patients participating in this study completed full lung func-tion testing, CT chest and CT pulmonary angiogram as well as cardiac MRI scan. Favourable opinion for this study was granted by the Ministry of Defence Research Ethics committee (1061/MODREC/20). All participants in the study provided written informed consent. The

**Table 1. Indications for referral to defence COVID-19 recovery service.**

| Criteria | |
|---|---|
| Patients with severe COVID-19 requiring hospitalisation | |
| Community patients with life-limiting symptoms more than 12 weeks after acute illness | |
| Patients desaturating to ≤95% oxyhaemoglobin saturation at rest or after 1 minute of the Harvard step test (30 steps per minute, 45 cm step) | |
| Chest pain and pathological ECG changes or rise in cardiac enzymes during the acute illness | |

remaining anonymised data are taken from the routine clinical care of patients attending the DCRS. In accordance with current policy surrounding the use of clinical data for education, evaluation and audit purposes, permission was granted for publication by the Defence Medical Services (DMS) Caldicott guardian and as such did not require formal ethical approval. All anonymised data was treated confidentially in accordance with Caldecott principles. The patient information and clinical data routinely collected in support of DCRS was used in this service evaluation. The data presented reflect the outcomes of every consecutive patient who attended the DCRS.

## Patient and public involvement

A patient engagement exercise was conducted in accordance with NIHR guidance prior to commencement of the DCRS clinical pathway and its associated research study: M-COVID (14). Four focus groups were conducted throughout July and August 2020 with 8–12 patients participating in semi-structured interviews. This resulted in amendments to written patient information with increased use of images and a co-ordinated effort across staff members to be consistent in messaging and (re)emphasis. In light of the cognitive symptoms affecting patients more time was allowed for explanations including an additional welcome briefing to re-iterate written information and allow for questions.

## Clinical review and investigations

Full clinical detail of the 3-day residential assessment pathway has been reported previously [7]. Initial assessment includes routine observations (body temperature, height, weight, blood pressure, heart rate, oxygen saturation and respiratory rate), 12-lead ECG, echocardiogram, blood tests, spirometry and a CPET (Fig 1).

## Patient symptoms and cognitive testing

All participants reported their acute and ongoing symptoms. Symptoms were recorded (present/absent) according to a pre-determined list of 37 symptoms. Standardised questionnaires were completed for breathlessness (modified BORG, 0–10 breathlessness scale); fatigue (fatigue assessment scale, FAS) [15,16]; anxiety (generalised anxiety disorder-7, GAD-7) [17], depression (patient health questionnaire 9, PHQ-9) [18], post-traumatic stress (posttraumatic stress disorder check list for DSM 5, PCL-5) [19] and alcohol consumption (alcohol audit) [20]. Population appropriate thresholds were employed for each questionnaire tool. For the FAS, >21 has been suggested as a threshold for 'substantial fatigue' [21] and >34 for 'extreme fatigue' [22]. The GAD-7 is a widely used screening tool for anxiety with a cut-off score of 10 identified to consider diagnosis based on a criterion standard study compared to independent mental health professional diagnosis [17]. The PHQ-9 is a commonly used screening tool for depression with a frequently used cut-off of 10 points to consider the diagnosis. Bivariate meta-analysis of 18 validation studies identified cut-off scores between 8–11 as optimal for

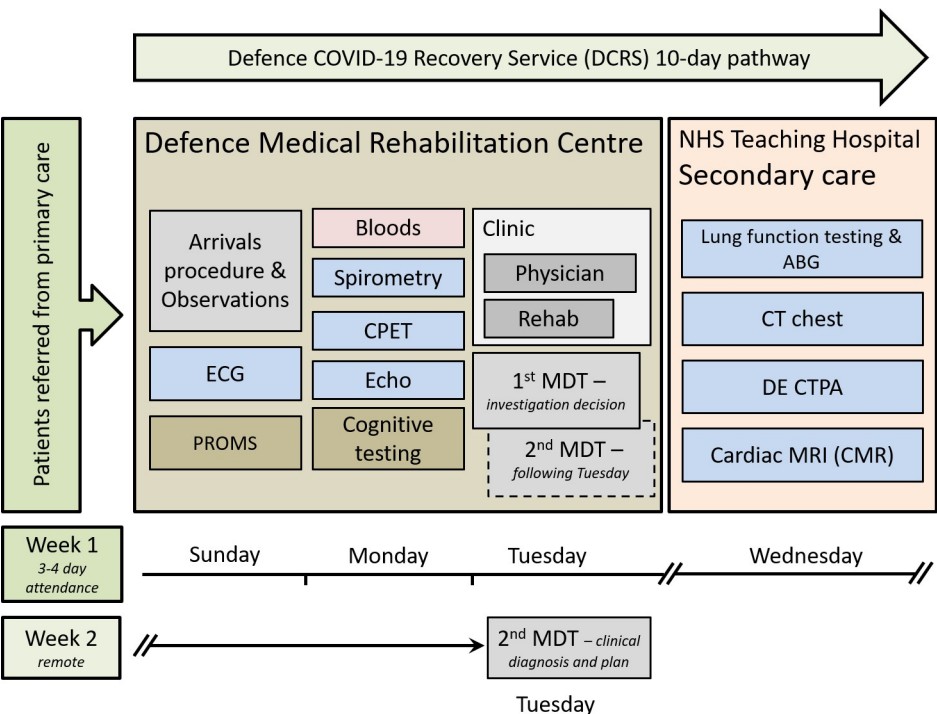

**Fig 1. Flow chart of the defence COVID-19 recovery service pathway.** ECG—electrocardiogram; PROMS–patient reported outcome measures, CPET–cardiopulmonary exercise test; MDT–multidisciplinary team meeting; ABG–arterial blood gas; CT High resolution computed tomography; DE CTPA–dual energy CT pulmonary angiogram; MRI–magnetic resonance imaging.

detecting major depressive disorder [18]. Signal detection analysis comparing the PCL-5 to the gold standard clinician administered PTSD scale indicated a score of 31–33 as an optimal cut-off for the diagnosis of PTSD in an US veteran population [19]. The EQ5D quality of life instrument was used to measure subjective wellbeing. This tool uses a visual analogue scale (0–100), similar to a thermometer, to record the participant responses [23,24].

Cognitive testing was undertaken using the National Institute of Health (NIH) cognition battery of the NIH Toolbox of Neurological and Behavioural Function (NIH-TB) which has been well-validated for this purpose [13]. A score more than 1.5 standard deviations below the mean was taken to represent a clinically significant deficit [25]. Cognition, which is one of four batteries, includes measures which map to several cognitive dimensions including executive function, episodic memory, language, processing speed, working memory and attention. Cognitive testing was supervised by trained personnel with neurological and psychology oversight and review of results. Testing was undertaken in a controlled environment. The output scores of cognitive testing permit the separation of cognitive performance into the 'fluid composite', which is recognised to be vulnerable to aging and biological insult and the 'crystallised composite' which is relatively preserved in the face of systemic disturbance and over the course of life [14]. In brief, the fluid component of cognition tends to deal with our current ability to react to, reason and deal with complex information whilst the crystallised component represents learning and knowledge acquired through life. The cognitive battery utilises a computer adaptive testing paradigm allowing a comprehensive assessment to be completed rapidly. The ordering of the testing disseminates tests mapping to both the above composite scores throughout the test period, thus participants are effectively blinded to the output. This allows objective determination of the fluid and crystallised scores.

## Clinician and patient grading of functional status

Functional status at the time of acute illness and at the point of follow up was agreed by the MDT based on the clinical history from the patient, the level of medical care required and the patient's status during the post-COVID clinic attendance. The WHO performance status (PS) 0–5 (Table 2) was used to record this functional performance [26]. Patients all reported their self-rated functional status using the functional activity assessment score (FAA) as fully fit [1]; fit for their working trade and for limited military duties [2]; unfit for their working trade but fit for limited military duties [3]; unfit for all but sedentary duties [4], or unfit for all duties [5].

## Observations, ECG, echocardiogram and bloods

Details of baseline observations, ECG and routine bloods are included in the S1 File.

## Spirometry and transfer factor/diffusing capacity

Forced vital capacity (FVC) and forced expiratory volume in the first second of expiration (FEV1) were measured in accordance with recommended guidelines [27] using Vyntus ONE Pulmonary Function testing equipment by Vyaire ™ Medical (Chicago Il, USA). Transfer factor for carbon monoxide (also termed diffusing capacity for carbon monoxide, DLCO) and alveolar volume (Va) measured using the tracer gas methane were measured over a ten-second breath hold. Results were adjusted for haemoglobin concentration [28].

## Cardiopulmonary exercise testing

Cardiopulmonary exercise test with capillary blood gas measurement at rest and stress was conducted on a stationary upright cycle ergometer (Corival, Lode, The Netherlands) and a Metalyzer3B (Cortex, Germany) calorimeter. A ramp protocol (10-35W/minute) was selected to achieve a 6–8 minute duration of loaded exercise. Full details are in the supplementary methods.

## Cardiothoracic imaging

High-Resolution CT chest (HRCT) and Dual-Energy CT Pulmonary Angiography (DECTPA) scans were acquired using a dual-source CT (Siemens SOMATOM Drive, Siemens Healthineers, Erlangen, Germany). CMR scans were performed on Siemens MR scanners at 1.5–3 Tesla (Siemens Medical Solutions, Erlangen, Germany). Details of scanning protocols are available in the supplementary methods.

**Table 2. WHO Performance Status (PS).**

| | WHO Performance Status (PS) Definition |
|---|---|
| 0 | Fully active, able to carry on all pre-disease performance without restriction |
| 1 | Restricted in physically strenuous activity but ambulatory and able to carry out work of a light or sedentary nature, e.g. light housework, office work |
| 2 | Ambulatory and capable of all self-care but unable to carry out any work activities. Up and about more than 50% of waking hours |
| 3 | Capable of only limited self-care, confined to bed or chair more than 50% of waking hours |
| 4 | Completely disabled. Cannot carry on any self-care. Totally confined to bed or chair |
| 5 | Dead |

## Parallel clinical rehabilitation service

Diagnostic and prognostic evaluation of workers with post-COVID illness was conducted in concert with a parallel rehabilitation pathway delivering remote (initial video tele-consultation) consultation with a rehabilitation consultant, followed by two week tailored residential rehabilitation programme where indicated.

## Baseline data

Baseline data for the denominator population (all UK regular service personnel) were taken from published UK government figures for the demography of the UK Armed Forces (UK armed forces biannual diversity statistics: index), or from specific information requests directed via Ministry of Defence Statistics.

## Statistical analysis

Statistical analysis was performed using GraphPad Prism (version 9; GraphPad Software Inc., La Jolla, CA, USA). Comparison of distinct groups was conducted using Mann-Whitney (non-parametric) or unpaired t-test (parametric) tests. Contingency table analysis was conducted by Fisher's exact test or $\chi^2$ test. Associations were tested using Spearman's correlation (and are reported with the Spearman r coefficient and p-value). Data are reported either by median (interquartile range), mean (±SD), or by number of cases, including the proportion (%) of the group studied and, where different, the proportion of the total number of patients assessed (%). A p-value of 0.05 (2-tailed) was taken to indicate statistical significance.

# Results

## Demographics

205 patients were assessed between August 2020 and April 2021. Our results are based on consecutive patients, with no exclusions. All are regular serving members of the British Armed Forces. The baseline characteristics of this group are reported in Table 3. Data are presented (where available) in parallel with the same characteristics for the whole Armed Forces population from which they are drawn. The group has a median age of 39 years (range 17–61; IQR 30–46.7 years), is predominantly male (83.4%) and overweight (median BMI 28.3, IQR 26–31.2; 33% are obese [BMI >30 kg/m$^2$]). 16.7% are of BAME ethnicity, compared to 9.1% in the Armed Forces population. The proportion of female patients is 16.6%, compared to the 11% of women in the UK Armed Forces. The majority have never smoked (72%) and the prevalence of cardiorespiratory disorders (asthma 7%, controlled Stage 1 hypertension 6%), metabolic disease (3%) and mental health diagnoses (10%) are low. 68% had laboratory-confirmed COVID-19 acute illness and the majority of those who did not, had clinically highly probable COVID-19 and suffered their acute illness during March and April 2020 when laboratory testing was not generally available. The median duration from acute illness to clinical assessment was 24 weeks (IQR 17.1–34 weeks).

## Symptoms

Most patients (69%) reported at least 3 ongoing symptoms, with only 16% symptom free. Among those treated in the community there was a larger proportion who remained symptomatic. Only 20/152 (13%) of community-treated patients were symptom free vs. 13/53 (25%) in the group who were hospitalised during acute illness. Table 4 lists the frequency of all symptoms which were reported by at least 10% of those assessed. S1 Table lists the relative frequencies of symptoms in those with severe acute illness requiring hospital admission, in comparison to those with mild/moderate acute illness managed in the community. Shortness

**Table 3. Baseline characteristics of DCRS patients.**

| | | Patient population | Armed Forces population |
|---|---|---|---|
| Number | | 205 | 146330 |
| Age | Mean age | 38.3 | 31.8 |
| | 10–19 | 4 (2%) | 8750 (6%) |
| | 20–29 | 44 (21.5%) | 57580 (39.4%) |
| | 30–39 | 65 (31.7%) | 50510 (34.5%) |
| | 40–49 | 63 (30.7%) | 23320 (15.9%) |
| Male Sex (%) | | 171 (83.4%) | 130230 (89%) |
| Ethnicity | White | 170 (83.3%) | 131930 (90.9%) |
| | BAME | 35 (16.7%) | 13,200 (9.1%) |
| BMI | | | |
| | >25 | 168 (82%) | 27% † |
| | >30 | 68 (33%) | 12.3% ‡ |
| Increased waist circumference | ♂ >102cm ♀ >88cm | 61 (30%) | |
| Smoking | | | |
| | current | 12 (6%) | |
| | ex-smoker | 46 (22%) | |
| | never smoked | 147 (72%) | |
| Medical History | Asthma | 14 (7%) | |
| | Hypertension | 12 (6%) | |
| | Diabetes | 3 (1.5%) | |
| | Pre-diabetes | 3 (1.5%) | |
| | COPD | 0 | |
| | Heart disease | 0 | |
| Pre-COVID psychiatric history | Anxiety | 5 (2.5%) | |
| | Depression | 13 (6%) | |
| | PTSD | 1 (0.5%) | |
| | other | 2 (1%) | |
| Current psychiatric diagnosis | Anxiety* | 11 (5.5%) | |
| | Depression* | 13 (6%) | |
| | PTSD* | 3 (1.5%) | |
| | Other* | 2 (1%) | |
| Lab testing | | | |
| | PCR +ve | 110 (53.7%) | |
| | AB +ve | 121 (59.0%) | |
| | PCR or AB +ve | 139 (67.8%) | |
| Acute illness | | | |
| | ED attendance | 92 (45%) | |
| | Hospital admission | 53 (25.9%) | |
| | ICU | 10 (4.9%) | |
| | I&V | 6 (2.9%) | |
| | Days ventilated | 14.5 [11.5–23.5] | |
| WHO Performance Status acutely | 0 | 0 | |
| | 1 | 25 (12.2%) | |
| | 2 | 92 (44.9%) | |

(*Continued*)

**Table 3.** (Continued)

| | | Patient population | Armed Forces population |
|---|---|---|---|
| | 3 | 78 (38%) | |
| | 4 | 10 (4.9%) | |
| | 5 | 0 | |
| Weeks from illness to DCRS review | | 24.0 [17.1–34.0] | |
| Symptomatic at review | | 172 (84%) | |

BAME: Black, Asian and minority ethnic; BMI: Body mass index (kg/m$^2$); COPD: Chronic obstructive pulmonary disease; PTSD: Post-traumatic stress disorder; PCR: Polymerase chain reaction; AB: Antibody (antispike AB test in unvaccinated, anti-nucleocapsid AB in vaccinated);ED: Emergency department; ICU: Intensive care unit; I&V: Intubated and ventilated; WHO: World Health Organisation; DCRS: Defence COVID-19 Recovery Service.

*There was no statistical difference between the prevalence of psychiatric disease pre- and post-COVID-19.

† Ministry of Defence–Lifestyles Steering Group. (2019). Data prepared by Defence Statistics.

‡ UK Armed Forces biannual diversity statistics. https://www.gov.uk/government/collections/uk-armed-forces-biannual-diversity-statistics-index.

of breath was the most frequent symptom, occurring in 125 (61%) of patients, with fatigue next, affecting 111 (54%). Cognitive symptoms impacting concentration, memory, and attention, and including confusion, affected nearly half of the group (47%). Symptoms of low mood, anxiety, and sleep disturbance were all described by more than one quarter of all patients. The likelihood of reporting ongoing cognitive symptoms was higher in community vs. hospitalised patients (S1 Table).

## Patient reported outcome measures

More systematic assessment of anxiety, depression, and post-traumatic stress markers, using the GAD-7, PHQ-9, and PCL-5 PROMS respectively, revealed that 17%, 24% and 13% of respondents met the threshold criteria used for consideration of referral to mental health services. Scores did not differ between hospitalised and community patients, though there was a trend toward higher depression score and lower quality of life score in community patients (S1 Table). The fatigue assessment scale tool provided an indication of those patients requiring further discussion with rehabilitation physicians, with 119 (58%) exceeding the threshold for referral. Fatigue scores were slightly higher in community vs. hospitalised patients (Fig 2).

## Cognitive testing

The NIH-TB cognitive battery was completed by every patient (Table 4). The fluid composite scores were lower than crystallised composite scores by a mean difference in T-score of 4.7 (p<0.001). A comparison of crystallised composite, fluid composite and total cognitive scores between patients who did, and did not, suffer with symptoms of fatigue, poor concentration, poor memory, low mood, and anxiety demonstrated that there was no significant between-group difference in crystallised composite scores, but a significant (p<0.05) between-group difference in fluid composite scores for all these symptoms (Fig 3). Cognitive scores did not differ significantly between community and hospitalised patients (S1 Table).

There are associations between the NIH-TB cognitive score and the scores for anxiety (GAD-7), depression (PHQ-9), post-traumatic stress (PCL-5) and fatigue (FAS) (Table 5). There are also significant associations between cognitive scores and self-reported symptoms of fatigue; low mood; anxiety; poor memory and poor concentration. These associations are driven by, and strongest within, those aspects of cognitive function which are grouped under the fluid composite score, which is recognised to be vulnerable to delirium and other forms of

**Table 4. Symptoms, Patient Reported Outcome Measures (PROMS) and cognitive scores.**

| Parameter | | Number (% of 205) | Abnormal/tested (%) |
|---|---|---|---|
| | Symptoms occurring at a frequency of ≥10% (n = 205) | | |
| No symptoms | | 33 (16%) | |
| Any shortness of breath | | 125 (61%) | |
| | On moderate activity | 50/125 (40%) | |
| | On mild activity | 47/125 (38%) | |
| | At rest | 28/125 (22%) | |
| Fatigue | | 111 (54%) | |
| Any cognitive symptoms | | 97 (47%) | |
| | Poor concentration | 82 (40%) | |
| | Poor memory | 63 (31%) | |
| | Poor attention | 51 (25%) | |
| | Confusion | 13 (6%) | |
| Muscle aches | | 63 (31%) | |
| Low Mood | | 58 (28%) | |
| Sleep | Difficulty getting to sleep | 58 (28%) | |
| | Difficulty staying asleep | 58 (28%) | |
| Anxiety | | 54 (26%) | |
| Exercise intolerance | | 50 (25%) | |
| Joint pain | | 48 (23%) | |
| Chest pain | | 47 (23%) | |
| Generalised weakness | | 43 (21%) | |
| Headache | | 42 (20.5%) | |
| Palpitations | | 41(20%) | |
| Dizziness | | 35 (17%) | |
| Loss of appetite | | 21 (10%) | |
| PROMS (n = 205) | | Median (IQR) | |
| GAD-7 anxiety | 0–21, consider MH referral >10 | 4 (2–7.5) | 34 (17%) n >10 |
| PHQ-9 depression | 0–27, consider MH referral >10 | 6 (3–10) | 50 (24%) n >10 |
| PCL-5 post-traumatic stress | 0–80, consider MH referral >31 | 9 (2–18) | 26 (13%) n >31 |
| EQ5-D | 0–100 (100 is maximum wellbeing score) | 70 (50–80) | |
| Fatigue assessment scale | 10–50, consider rehab referral >21, >34 is severe | 24 (18.5–30) | 119 (58%) n>21 |
| Alcohol Audit | 0–40, >7 likely harm | 4 (2–5) | 26 (13%) n>7 |
| Cognitive Scores (n = 205) | | | |
| Crystallised composite | T-score (50 represents population median) | 57 (50–65) | 4 (1.9%) >1.5SD† below the mean |
| Fluid composite | | 53 (43–60) | 17 (8.3%) >1.5SD† below the mean |
| Total composite | | 56 (49–63) | 7 (3.4%) >1.5SD† below the mean |

PROMS: Patient reported outcome measures; MH: Mental health; GAD-7: Generalised anxiety disorder-7 (17); PHQ-9: Depression module of the patient health questionnaire (18); PCL-5: Post-traumatic stress disorder checklist for DSM-5 (diagnostic and statistical manual of mental disorders) [18]; EQ5-D: EuroQoL 5-dimension quality of life instrument [23]. Fatigue assessment scale thresholds of severity >21 'substantial' and >34 'severe' [21,25].

†1.5 standard deviations below the mean has been used as the cut-off for significant impairment, in keeping with accepted practice within behavioural science [25].

biological insult. No significant association exists between the crystallised component of cognitive performance and any of the patient reported outcome measures of psychiatric illness, emotional distress or cognitive symptoms. By contrast, there is no relationship between cognitive score and any aspect of lung function testing; pathology on CT chest imaging or cardiac

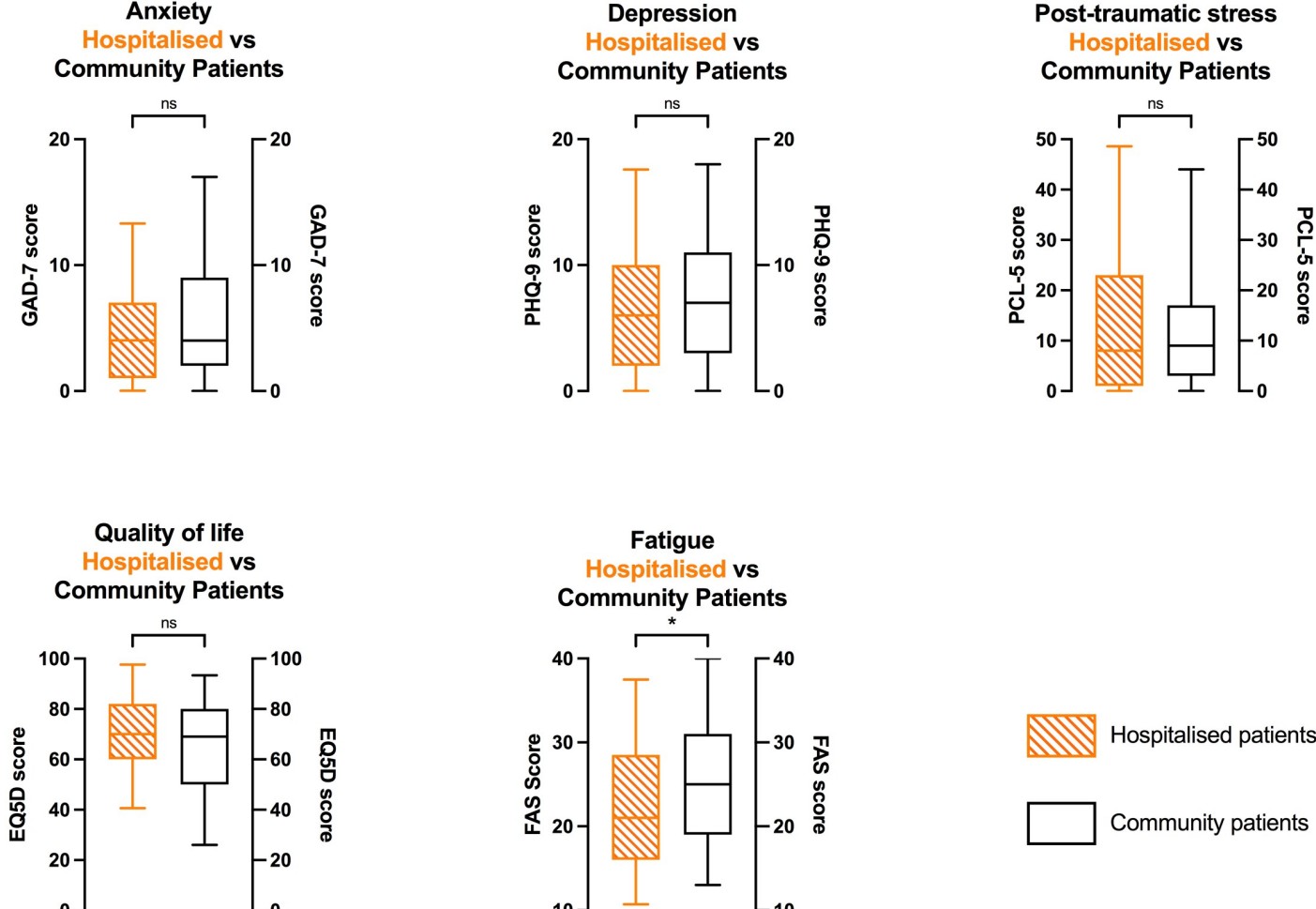

**Fig 2. Comparison of patient reported outcome measures (PROMS) between severe acute COVID-19 illness (defined by requiring hospitalisation) and COVID-19 treated in the community.** GAD-7: Generalised anxiety disorder-7 [0–21]; PHQ-9: Depression module of the patient health questionnaire [0–27]; PCL-5: Posttraumatic stress disorder checklist for DSM-5 (diagnostic and statistical manual of mental disorders) [0–80]; EQ5-D: EuroQoL 5-dimension quality of life instrument [0–100]; FAS: Fatigue assessment scale [10–50].

MRI scan. In terms of the relationship between cognitive performance and objective functional capacity, there is a modest association between both total cognitive score, and fluid composite score, and the peak predicted $VO_2$ (both Spearman r 0.15, p<0.05). No association exists between functional capacity and the crystallised composite.

## Performance status

WHO Performance status (PS) and FAA scores are reported in S2 and S3 Tables. There was a significant improvement in median functional status from a median PS of 2 during acute illness to 1 (Table 2) at DCRS assessment. However, 67% had not returned to their pre-COVID function and only 21% rated themselves as 'fully fit'. Neither physician graded PS nor self-rated functional activity assessment differed between hospitalised and community patients (S1 Table). Although there was no significant association between PS and cognitive scores, there was a trend (p = 0.07) towards lower fluid composite scores (reduction in mean fluid composite T score of 8.3) for those with a performance status of 2 (reduced performance)

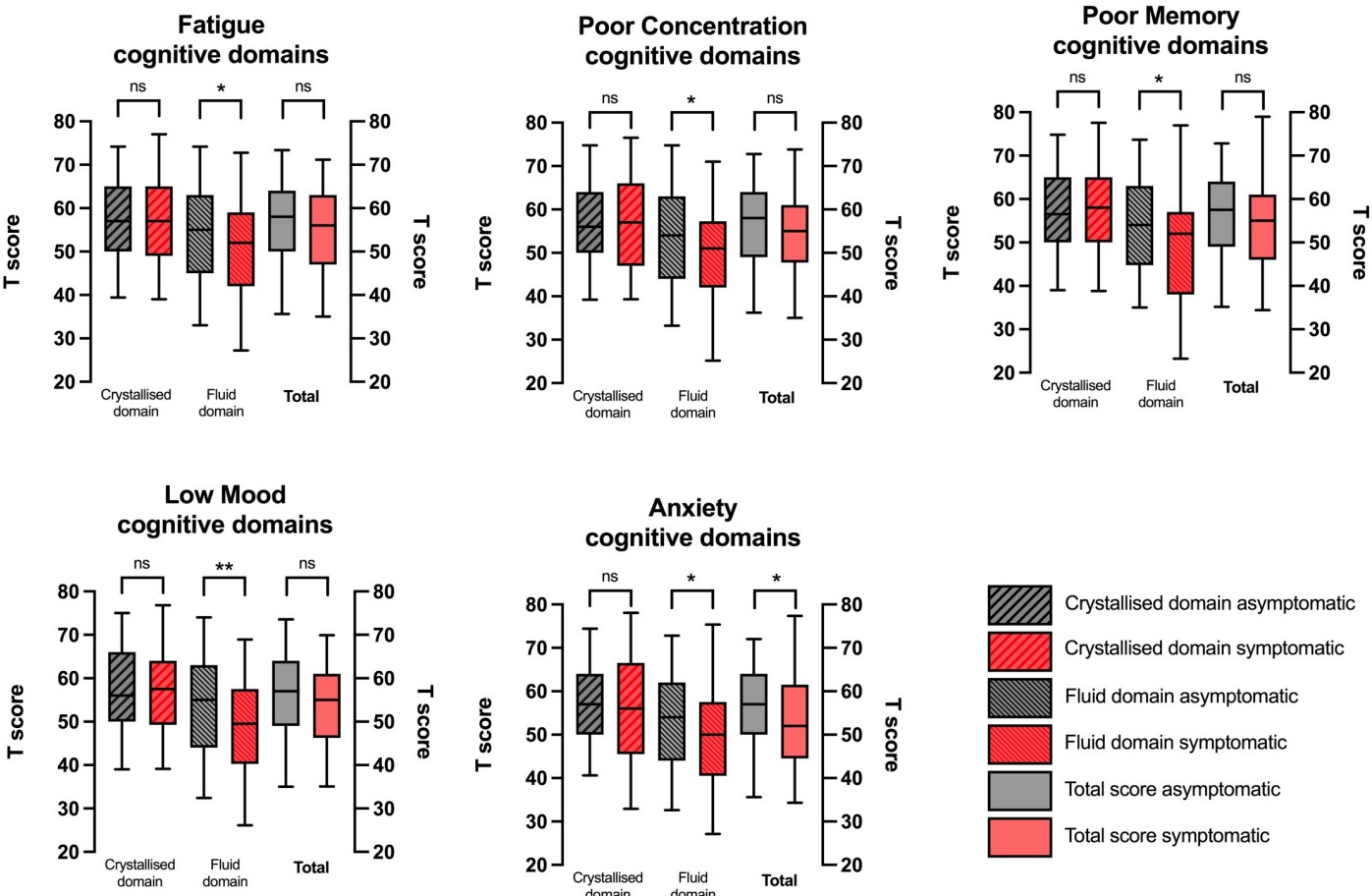

**Fig 3. Comparison of fluid and crystallised cognitive domain scores between those with vs. without symptoms of cognitive dysfunction and mood disorder.** Cognitive scores were all obtained using the standardised NIH-TB cognitive battery. The differences indicated refer to a statistically significant numerical difference between the group scores for patients with an ongoing symptom vs. those without * p<0.05; **p<0.01.

compared to a PS of 0 (normal). There was no between-group difference in total cognitive score (p = 0.49) or in crystallised component (p = 0.34) values. There were modest associations between reduced performance status and scores for anxiety [GAD-7] (Spearman r 0.318, p<0.0001); depression [PHQ9] (Spearman r 0.425, p<0.0001); post-traumatic stress [PCL-5] (Spearman r 0.34, p<0.0001); fatigue [FAS] (Spearman r 0.48, p<0.0001); and symptoms of poor memory, poor concentration and poor attention (Spearman r values 0.31, 0.32 and 0.23, all p<0.0001). There was no relationship between performance status and likelihood of identifying abnormalities on CT or on CMR, but there was, a weak association between worse performance status and lower FEV1 (Spearman r -0.2, p<0.01). More predictably, there was an association between worse performance status and decreased cardiopulmonary fitness [peak $VO_2$ (l/min)] (Spearman -0.3, p<0.0001) and ventilatory inefficiency [VE/VCO_2, higher value signifies worse efficiency] (Spearman 0.35, p<0.0001).

## Blood results

Routine blood test data are summarised in S4 Table. 21% of estimated glomerular filtration rate (eGFR) values are below 90 ml/min, though, only 1% of patients had a creatinine above the upper

**Table 5. Cognitive score: Association with PROMS, symptoms, medical findings and functional capacity.**

| Parameter | | Fluid component | | Crystallised component | | Total cognitive score | |
|---|---|---|---|---|---|---|---|
| | | Spearman r | p-value | Spearman r | p-value | Spearman r | p-value |
| PROMS | GAD-7 | -0.25 | p<0.0001 | -- | ns | -0.21 | p<0.01 |
| | PHQ-9 | -0.24 | p<0.001 | -- | ns | -0.19 | p<0.01 |
| | PCL-5 | -0.29 | p<0.0001 | -- | ns | -0.25 | p<0.0001 |
| | FAS | -0.23 | p<0.001 | -- | ns | -0.20 | p<0.01 |
| Symptoms | Anxiety | -0.14 | p<0.05 | -- | ns | -0.14 | p<0.05 |
| | Low mood | -0.20 | p<0.01 | -- | ns | -0.14 | p<0.05 |
| | Fatigue | -0.15 | p<0.05 | -- | ns | -- | ns |
| | Poor memory | -0.15 | p<0.05 | -- | ns | -0.14 | p<0.05 |
| | Poor concentration | -0.14 | p<0.05 | -- | ns | -- | ns |
| Lung function | FEV1%predicted | -- | ns | -- | ns | -- | ns |
| | FVC %predicted | -- | ns | -- | ns | -- | ns |
| | Alveolar volume %predicted | -- | ns | -- | ns | -- | ns |
| | DLCO %predicted | -- | ns | -- | ns | -- | ns |
| | KCO %predicted | -- | ns | -- | ns | -- | ns |
| CT chest pathological findings | | -- | ns | -- | ns | -- | ns |
| Cardiac MRI pathological findings | | -- | ns | -- | ns | -- | ns |
| Cardiorespiratory fitness | % predicted Pk $\dot{V}O_2$ | 0.15 | p<0.05 | -- | ns | 0.15 | p<0.05 |

PROMS: Patient reported outcome measures; GAD-7: Generalised anxiety disorder-7 [17]; PHQ-9: Depression module of the patient health questionnaire [18]; PCL-5: Post-traumatic stress disorder checklist for DSM-5 (diagnostic and statistical manual of mental disorders) [19]; FAS: Fatigue assessment scale. FEV1%predicted: Forced expiratory volume in 1s, % of the predicted value; FVC: Forced vital capacity; DLCO: Diffusing capacity for carbon monoxide; KCO: Carbon monoxide transfer coefficient; CT: Computed tomography; MRI: Magnetic resonance imaging; pk $\dot{V}O_2$: Uptake of oxygen at peak exercise.

limit of normal (ULN). Importantly, CRP values did not suggest persistent systemic inflammation. Median CRP was 1.2 (0.7–2.1) mg/L, with only one value above the ULN (10 mg/L) in a patient identified as having acute COVID-19 pneumonitis, in spite of negative PCR prior to attending DCRS. 17% of ALT measurements and 13% AST measurements were above ULN.

## Lung function testing

Lung function testing was normal in all respects in two thirds of the 205 patients assessed. Alveolar volume and transfer factor ($D_{LCO}$) were reduced in 13% and 16% respectively. Reduced transfer coefficient ($K_{CO}$) occurred in only 3%. In this population, despite a high proportion of obese patients, the forced vital capacity was reduced in only 4%. FEV1 was reduced in 11%, with a reduced FEV1/FVC ratio (less than 0.70) in 13%. There was no difference between lung function testing parameters in hospitalised vs. community patients, excepting a small difference in FVC (99% (94–109%) vs. 106% (98–117%), p = 0.005).

## Physical performance

Functional capacity, measured by CPET, showed that median peak work was 96.9% (88–111%) predicted and that median peak oxygen uptake (peak $\dot{V}O_2$) was 109% (100–125%) predicted. 21% and 6% respectively of patients assessed failed to reach 85% of the peak predicted

work and peak predicted oxygen uptake. 11% of patients had a $\dot{V}E/\dot{V}CO_2$ ratio at anaerobic threshold (indicator of ventilatory efficiency) above 30.0 (showing suboptimal ventilatory efficiency).

## Imaging

Clinical investigation results are reported in Table 6. Every patient had spirometry and CPET. Dual-energy CT pulmonary angiography, CT chest and cardiac MRI scans were completed in 80, 97 and 90 patients respectively. All cross-sectional imaging data are included.

## Lung HRCT

HRCTs were clinically indicated in 47% of patients. They were abnormal in 31/97 (32%) of the scans acquired. 76% of the abnormalities identified were attributed to COVID-19, including persisting ground-glass changes, fibrosis, or a combination of these, representing 11% of the total patient group. CT changes attributable to COVID were identified in 19/40 (48%) scans in those hospitalised during their acute infection and in 5/57 (9%) scans in community patients, a 5.4-fold increased likelihood in those hospitalised during their acute illness (p<0.0001). The negative predictive value of not having required hospital admission on finding COVID-19 related lung pathology on HRCT is 91%. Two CTs were reported to show changes of 'moderate' severity. Both of these had persisting ground glass change; affecting 25–50% and 50–75% of lung volume respectively. A third HRCT revealed acute COVID-19 pneumonitis. In all other abnormal HRCTs (28 scans) the changes were reported as 'mild' or 'very mild'.

## Dual energy CTPA

Only 2/80 (3%) dual energy DECTPAs showed a pulmonary vascular abnormality. One showed subsegmental pulmonary embolus and the other a pattern consistent with high likelihood of multiple subsegmental pulmonary emboli. In both cases the likelihood that these findings related to COVID pneumonitis was very high.

## Echocardiogram

An echocardiogram was acquired in the first consecutive 113 patients. At this point, following a review of the clinical findings, echocardiography was removed from the pathway based on not identifying sufficient pathology to justify its continuation. A case of mild aortic regurgitation and another of mildly elevated estimated pulmonary artery systolic pressure (31 mmHg + right atrial pressure of 5–10 mmHg) were identified. In neither case were the findings associated with ventricular dilatation or dysfunction.

## Cardiac MRI

Abnormalities were reported in 29/90 (32%) CMRs, with 13 in the hospitalised group. 15 of 29 (52%) abnormal scans described borderline/mild reduction in LV systolic function ('Grey zone' left ventricular ejection fraction 50–56% on cardiac MRI [29]), typically associated with borderline LV cavity dilatation. Among this group of 15 patients, the majority had a peak $\dot{V}O2$ at or above 100% predicted. 4 had peak $\dot{V}O2$ >110% predicted and a peak $\dot{V}O2/HR$ (correlated with cardiac stroke volume) > 120% predicted. 7 CMR scans showed LV wall thickening consistent with hypertensive change. Previous/resolving myocarditis was identified in 7 (8%) patients investigated with CMR. This represents 3.5% of all patients assessed. 4 cases occurred in hospitalised patients and 3 in non-hospitalised patients. In no case of myocarditis was there

**Table 6. Full results of clinical investigation.**

| Parameter | | | Value (IQR) | Abnormal/tested (%) | Abnormal /total (%) |
|---|---|---|---|---|---|
| Lung function (spirometry all) | | | | | |
| FEV1 | | | 3.89 (3.32–4.33) | | |
| FEV1% predicted | | | 98% (89–106) | 22/205 (11%) <80% pred | 11% |
| FVC | | | 5.08 (4.29–5.69) | | |
| FVC % predicted | | | 104% (96%-116%) | 9/205 (4%) <80% | 4% |
| FEV1/FVC ratio | | | 0.77 (0.74–0.80) | 26/205 (13%) <0.70 | 13% |
| $D_{LCO}$ % predicted | | | 85 (77–93.5) | 33/95 (35%) | 16% |
| $K_{CO}$ % predicted | | | 98 (92–108.5) | 6/95 (6%) | 3% |
| Alveolar volume | | | 5.68 (4.72–6.43) | | |
| Alveolar volume % predicted | | | 86% (78–94) | 26/95 (27%) <80% | 13% |
| any abnormal lung function test | | | | 67/205 (33%) | 33% |
| CPET (all) | | | | | |
| RER | | | 1.1 (1.1–1.2) | | |
| Work (W) | | | 240 (213–270) | | |
| Work % predicted | | | 96.9% (87.8%-110.5%) | 44 (21%) <85% pred | 21% |
| V̇O2 l/min | | | 3.1 (2.6–3.4) | | |
| V̇O2 % predicted | | | 109.3% (100%-124.9%) | 12 (6%) <85% pred | 6% |
| V̇E/V̇CO2 at AT | | | 26.3 (24.4–28.2) | 23 (11%) >30.0 | 11% |
| CT Chest (97 scans) | | | | | |
| Abnormal CT | | | | 31/97 (32%) | 15% |
| Abnormal CT in hospitalised | | | | 23/53 (43%) | -- |
| COVID changes | | | | 24/97 (25%) | 12% |
| | ground glass and fibrosis | | | 9/97 (9.5%) | 4.5% |
| | ground glass only | | | 11/97 (11.5%) | 5.5% |
| | fibrosis only | | | 4/97 (4%) | 2% |
| Bacterial pneumonia | | | | 3 (3%) | 1.5% |
| Other | | | | 4 (4%) | 2% |
| Dual energy CT pulmonary angiogram (80 scans) | | | | | |
| Abnormal CTPA | | | | 2/80 (2.5%) | 1% |
| | | 1 case of subsegmental embolus, 1 of multiple microemboli | | | |
| Cardiac MRI (90 scans) | | | | | |
| Abnormal CMR | | | | 26/90 (29%) | 12.5% |
| Abnormal CMR in hospitalised | | | | 13/53 (25%) | -- |
| | Myocarditis | | | 7/90 (8%) | 3.5% |
| | Borderline/mildly reduced LV systolic function 'Grey zone' | | | 15/90 (17%) | 7% |
| | Mild LVH | | | 4/90 (4%) | 2% |

an abnormality on the 12-lead ECG in clinic, reduced systolic function or regional wall motion abnormality at the time of CMR. The frequency of occurrence of myocarditis did not differ (p = 0.7) between hospitalised and non-hospitalised patients.

### Effect of time from acute illness to assessment

There was no correlation between time from acute illness to assessment for any symptom or any of the PROMS (anxiety, depression, post-traumatic stress, quality of life). There was also no correlation between time to assessment and the likelihood of finding pathology on CT chest imaging or cardiac MRI scan. There was, however, a modest correlation between a small improvement in peak oxygen uptake on CPET (Spearman r = 0.189, p <0.01) and on a small improvement in FEV1 and FVC (Spearman r = 0.17 and 0.2 respectively, both p <0.05).

## Discussion

Despite low rates of residual cardiopulmonary pathology in this young active cohort, with very low rates of premorbid illness, a high burden of symptoms remained and a large proportion had not recovered pre-COVID function when assessed 6-months after acute illness. One third had not returned to their pre-COVID functional status (physician rated WHO performance status) and only one in five self-rated as 'fully fit'. This persistent impairment disproportionately impacted those managed in the community. These functional limitations were similar to a comparable UK post-hospital cohort (PHOSP-COVID) of ~1,000 middle aged UK citizens (mean age 58, 36% female, 2/3 working at the time of acute illness) at 5.9 months post hospital discharge. In this group only 29% of patients felt fully-recovered. In our study, systematic cognitive assessment identified specific deficits, which appear to contribute significantly to the symptomatology of long-COVID.

## Limitations

The principal limitation of this work, namely that it reflects a tightly defined, predominantly male, Armed Forces population, is also linked to a key strength. This population is comparatively young, comparatively fit at pre-COVID baseline (annual fitness testing is compulsory), in full-time work, is drawn from across all socioeconomic groups in the UK and has a low prevalence of comorbid illness. The baseline characteristics of the population are well defined and accurately recorded. The DCRS provides a comprehensive, uniform clinical assessment in a single, common pathway, to assess those individuals from the ~150,000 UK Armed Forces population who are significantly impacted by COVID-19. This presents an invaluable opportunity to measure the effect of COVID-19 in a working-age population. In addition, the delivery of a uniform, comprehensive evaluation of physical, neurocognitive and clinical endpoints, by chest physicians, cardiologists, radiologists, neurology and rehabilitation specialists, in both the DMS and a single NHS Trust, allows assessment with few confounding factors. Although there was a small association between longer time from acute illness to assessment and improved peak cardiorespiratory fitness (peak $\dot{V}O_2$ on CPET), and also with a slightly higher FEV1 and FVC, it is not possible to determine to what extent this is a 'recovery with time' effect or a reflection of referral bias, with sicker patients being referred to the service more promptly.

The predominantly male population seen in the DCRS are older and more overweight than the baseline military population. Although BMI is not necessarily a good predictor of percentage body-fat in a young, athletic group (owing to the high mass of skeletal muscle, which also probably accounts for the high proportion estimated to have low GFR), nearly a third of those seen had waist circumference in the very high-risk range [30,31]. BAME personnel and women are both over-represented compared to the baseline Armed Forces population with a low prevalence of premorbid illness or mental health diagnosis in the group. At the time of assessment, 6 months post-acute illness, there remains a high prevalence of (usually multiple)

symptoms, principally shortness of breath, fatigue, cognitive symptoms, low mood, and anxiety, especially in community managed vs. hospitalised patients. Although there is a significant improvement in WHO performance status between acute illness and the assessment six months later, two-thirds of patients had not returned to their pre-COVID performance status.

Although an abnormality in lung function testing was seen in one third of patients, this comprised a group with mild obstructive airways disease and another with mildly reduced alveolar volume associated with reduced transfer factor, suggesting a small reduction in alveolar volume rather than a diffusion abnormality. HRCT imaging demonstrated low levels of COVID-related pathology in our cohort, with a five-fold increased likelihood of abnormalities in patients hospitalised by COVID-19. Importantly, even in those with HRCT findings, there is a relatively limited extent of parenchymal disease in the majority and even in those with disease, ventilatory limitation was not a common finding on CPET. Furthermore, the very mild degree of functional limitation identified by CPET is unlikely to fully explain the common symptom of breathlessness. Pulmonary thromboembolic disease was very uncommon, affecting <1% of the cohort. Myocarditis was also a relatively uncommon finding (8% of CMRs), with no cases associated with ventricular dysfunction or regional wall motion abnormality. Most of the abnormalities identified by CMR were of borderline reduced systolic function. The robustly normal/supranormal CPET findings in this group suggest that the majority of these are likely to be explained by exercise adaptation rather than heart muscle disease.

Despite the relative lack of cardiopulmonary pathology, there remains a high prevalence of persistent symptoms including breathlessness, fatigue and various descriptions of cognitive impairment, that are more common in the community patients, compared to hospitalised patients who experienced more severe acute disease. Systematic cognitive testing in all patients a median of 6 months post illness allows us to report, for the first time, that patients with symptoms related to cognitive problems and mood disorder have objective focal cognitive deficits which affect those cognitive processes known to be vulnerable to ageing and intoxication [32]. Participants in this study had a reduction in fluid composite T-score of 4.7, p = 0.001, this is equivalent to a 7-point reduction on a standard IQ test and is of similar magnitude to the reduction seen in cognitive performance whilst intoxicated at the UK/US drink driving limit (80 mg/100 mL blood or 35 mcg/100 mL breath) or that seen with 10 years of normal ageing [32,33]. Of note, the mean drink drive limit for alcohol in Europe is slightly lower at 50 mg/ 100 mL blood. There is an association between cognitive scores and both patient reported outcome measures and symptoms of cognitive impairment, anxiety and low mood. This link, present in the fluid composite but not the crystallised composite of the cognitive score, was not seen with medical findings of lung function, CT chest imaging or cardiac MRI. Whilst not conclusive, these findings suggest that long COVID is a syndrome resulting from a prolonged impact upon neurocognitive function rather than cardiopulmonary organ dysfunction. This was also a key conclusion of the PHOSP-COVID investigators. They found no clear association between the severity of acute illness and likelihood of prolonged symptoms. An important difference between our own study and the PHOSP-COVID group was in the association of cognitive impairment with mental and physical illness. Whilst the PHOSP-COVID group identified correlations between symptoms, mental health measures and physical health across four clusters of disease severity, they found that cognitive scores were 'independent'. Cognitive impairment in PHOSP-COVID was assessed using the Montreal cognitive assessment (MoCA) [34]. Whilst MoCA has out-performed the mini mental state examination in detection of mild cognitive impairment in dementia screening, its validation is based in an elderly population. The cited reference in their report is a validation in patient and control groups with a mean age of 73–77 years. It is possible that the MoCA is insensitive to discriminate small but important changes in cognitive function in a working-age population.

Symptoms of fatigue and 'brain-fog' are non-specific and are seen in the context of a global pandemic causing understandable health anxiety, which might be expected to contribute to them [9]. Sensitive standardised cognitive testing, appropriate to the age-group assessed, raises the possibility of a better-defined, objective measurement of 'long COVID', which is currently a diagnosis of exclusion. Just as cardiopulmonary exercise testing may provide a means to rule out functionally significant lung or heart disease, systematic cognitive testing may offer a tool to 'rule in' objective neurocognitive insult in the wake of this prevalent disease. However, repeat longitudinal testing is required to confirm either persistence or resolution of cognitive symptoms and deficits.

This study recapitulates the finding of increased representation of high BMI, BAME ethnicity and female sex from previous studies [7,35], now seen alongside performance status and occupational impact in this young, fit, working population with few comorbid diagnoses. It provides reassurance regarding the low likelihood of lung pathology in those who did not require admission to hospital and very low frequency of clinically significant heart disease in contrast to earlier reports [36,37] and in keeping with more recent data [38,39].

Two important questions remain unanswered. First, how long symptoms last for most patients and second, what is the most effective approach to managing those with long-COVID. A large proportion of the DCRS patients volunteered for a prospective observational cohort study to investigate the longer-term effects of COVID-19 in the Defence population (M-COVID study: Ministry of Defence Research Ethics committee (1061/MODREC/20)) which may help answer the first, there is still much we need to understand before we can confidently answer the second.

## Conclusion

Six months after acute COVID-19 illness, despite a low frequency of cardiopulmonary pathology, a young, comparatively fit cohort, in full-time employment, continue to experience high rates of persistent symptoms; demonstrable cognitive impairment, akin to ageing by ten years, and the majority have not regained their pre-COVID function.

## Supporting information

**S1 Checklist.**
(DOCX)

**S1 Table. Common symptoms, patient reported outcome measures, cognitive testing scores and functional status in patients hospitalised by acute illness vs. managed in the community.**
(DOCX)

**S2 Table. WHO Performance Scale (PS) acutely and at DCRS clinic assessment.**
(DOCX)

**S3 Table. Self-rated Functional Activity Assessment (FAA) at DCRS.**
(DOCX)

**S4 Table. Additional clinical investigation results.**
(DOCX)

**S1 File. Additional detail on clinical assessments and investigations not given in the main text methods.**
(DOCX)

## Author Contributions

**Conceptualization:** David A. Holdsworth, James L. Mitchell, Nick P. Talbot, Alexander N. Bennett, Edward D. Nicol.

**Data curation:** David A. Holdsworth, Rebecca Chamley, Rob Barker-Davies, Oliver O'Sullivan, Peter Ladlow, Edward Sellon, Joseph Mulae.

**Formal analysis:** David A. Holdsworth, Cheng Xie.

**Investigation:** Rebecca Chamley, Rob Barker-Davies, Oliver O'Sullivan, Peter Ladlow, Dominic Dewson, Daniel Mills, Samantha L. J. May, Cheng Xie, Joseph Mulae, Oliver J. Rider.

**Methodology:** David A. Holdsworth, Alexander N. Bennett, Edward D. Nicol.

**Project administration:** David A. Holdsworth, Rebecca Chamley, Rob Barker-Davies, Oliver O'Sullivan, Mark Cranley, Edward Sellon, Joseph Mulae, Jon Naylor, Alexander N. Bennett, Edward D. Nicol.

**Resources:** David A. Holdsworth, Rob Barker-Davies, Oliver O'Sullivan, Alexander N. Bennett, Edward D. Nicol.

**Supervision:** David A. Holdsworth, Rebecca Chamley, Rob Barker-Davies, Joseph Mulae, Jon Naylor, Oliver J. Rider, Alexander N. Bennett, Edward D. Nicol.

**Visualization:** David A. Holdsworth.

**Writing – original draft:** David A. Holdsworth, James L. Mitchell.

**Writing – review & editing:** David A. Holdsworth, Rebecca Chamley, Rob Barker-Davies, Oliver O'Sullivan, Peter Ladlow, Joseph Mulae, Betty Raman, Nick P. Talbot, Oliver J. Rider, Alexander N. Bennett, Edward D. Nicol.

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
