## [Decision Letter · Decision Letter 0]

22 Dec 2021

PONE-D-21-34299Comprehensive clinical assessment identifies specific neurocognitive deficits in working-age patients with long-COVID.PLOS ONE

Dear Dr. Holdsworth,

Thank you for submitting your manuscript to PLOS ONE. After careful consideration, we feel that it has merit but does not fully meet PLOS ONE’s publication criteria as it currently stands. Therefore, we invite you to submit a revised version of the manuscript that addresses the points raised during the review process.

We look forward to receiving your revised manuscript.

Kind regards,

Lucette A Cysique, PhD

Academic Editor

PLOS ONE

Journal Requirements:

Additional Editor Comments :

Thank you for addressing the reviewers' comments

Reviewers' comments:

Reviewer's Responses to Questions

**Comments to the Author**

1. Is the manuscript technically sound, and do the data support the conclusions?

Reviewer #1: Yes

Reviewer #2: No

2. Has the statistical analysis been performed appropriately and rigorously? 

Reviewer #1: Yes

Reviewer #2: Yes

3. Have the authors made all data underlying the findings in their manuscript fully available?

Reviewer #1: Yes

Reviewer #2: No

4. Is the manuscript presented in an intelligible fashion and written in standard English?

Reviewer #1: Yes

Reviewer #2: Yes

5. Review Comments to the Author

Reviewer #1: The authors present results from a single-center observational study of 205 consecutive patients ca. 6 months after probable to secured SARS-CoV-2 infection, with the aim of identifying deficits compatible with a post-COVID syndrome. They used a comprehensive clinical assessment with focus on cognitive, cardiac and pulmonary function, using both objective measures and PROMs. Although the characteristics of the sample may somehow limit generalization, which is correctly discussed by the authors as a possible limitation, their homogenous setting assures a comprehensive assessment with few missing data. The results are nonetheless in line with previous evidence of frequent prolonged symptomatology even in less severe cases of SARS-CoV-2 infection and in absence of significant pre-morbid conditions.

Introduction

1. The introduction would benefit from referencing more recent publications with similar aims originating from patients with confirmed infection.

Methods

2. Although ethical review board approval is mentioned, it is not clear if/ how participants were informed about and consented study participation, particularly given that at least one participant was assumedly underage (age rage 17-61).

3. Were the symptoms reported based on a priori defined listing of possible symptoms or categorized after data was collected?

Results

4. Was there an association between the different measures and time after infection?

5. Table 3: It would be interesting to check if frequency of psychiatric history significantly differs of pre- vs post-COVID (current)?

6. Table 3: Table legend should be revised and all abbreviations should be explained

7. Was there information regarding timing of symptoms? I.e., do they represent ongoing symptoms since infection or appeared after the acute phase?

8. Table 4: It is not stated what MH stands for and it could be clarified on which guidelines the associated cut-offs (GAD-7, PHQ-9, PCL-5, Fatigue) are based on.

9. It should be clarified that the value given for EQ5D is the visual analog scale (VAS) and no index values are presented, which would be preferable.

10. For the results presented in Figure 3, it is not clear how cognitive dysfunction was defined (e.g., based on a priori set of deficits or associated severity, by domain/test).

11. It would be interesting to check if performance status (WHO Performance status and/or FAA score) is associated with other available measures, namely cognitive dysfunction, fatigue, lung and cardiac function.

Discussion

12. Similar to the introduction, the manuscript would benefit if authors relate their results with recently published data on similar groups.

Reviewer #2: Thus, the manuscript under review is a very important since it shows chronic health disorders associated with COVID-19 in previously healthy young or middle-aged people. The cohort consists of 205 employees of military institution, mainly men. The subjects were thoroughly tested in terms of somatic functions. A standardized set of neuropsychological tests was used to study cognitive function (NIH-TB). The results show that in a significant proportion of participants, symptoms of fatigue, cognitive and emotional disturbances persist several months after COVID-19, even if the disease was not severe and there are no significant pathological changes in organs such as the lungs or the heart.

Despite my initial enthusiasm, the manuscript raises many doubts, as below.

• The title of the manuscript and the actual content

By reading the title of manuscript, the reader is looking forward to results showing the relationship between medical findings and post-Covid-19 neurocognitive impairment. Certain medical indices are expected to be significantly related to the state of cognition. There is no such data presented. The data from the medical measurements and the results of the NIH-TB battery test are reported independently to characterize the number of participants with abnormalities. It is not known if and what medical data or other variables (fatigue, emotional disturbance) may indicate cognitive impairment after COVID-19 in this group. The key would be either to present these analyzes or to change the title.

• Cognitive disorders and reference measures in conclusions

There are no data and analyzes that would justify referring these cognitive disorders to the disorders detected in drunk drivers (according to the alcohol consumption limit for the UK) or the reduction of cognitive functions to the level typical for people 10 years older. Perhaps these comparisons are fully justified, but in order to draw such conclusions, the authors should refer to relevant publications or analyze the results of NIH-TB performance by drunk drivers or the standardized NIH-TB results for a group 10 years older.

By the way, what is the UK drink driving limit? Why the readers are supposed to know it?

• Measurement of cognitive functions and data analysis

The authors reduce the results of the NIH-TB to two components: crystallized and fluid. They define the fluid component as "ability to ... deal with complex information", the crystallized component: as learning and knowledge acquired through life. The definitions are not clear since the tasks involving crystallized functions may also involve dealing with complex information.

Moreover, the division into crystallized and fluid components is quite archaic and tells little about complex cognitive functions such as attention, executive functions, learning and memory, spatial orientation, information processing speed, and language.

The need for a detailed analysis of the results of the cognitive function examination can be illustrated by the following example. A reduction in the efficiency of NIH-TB was found, mainly in the "fluid component" but not the "cristalised one". Language abilities depending on education, lifelong learning are usually included in the crystallized component of intelligence. However, neuropsychological studies on people after COVID-19 show that at least some aspects of language functions (naming, verbal fluency, semantic abilities) may be impaired in an effect on COVID-19. How, then, to understand the preserved vs. impaired abilities in the studied group? The results of available neuropsychological studies on COVID-19 should be presented in detail in the Introduction part and carefully discussed in the Discussion section.

• Use of abbreviations

Text and tables in many places are confusing by using abbreviations without explanation. Also, the use of abbreviation “HRCT” (high resolution CT) without specifying the organ (lungs, heart, brain) is completely incomprehensible (see Abstract).

• Other comments

Were neurological and neuroimaging examinations included in this comprehensive medical assessment, esp. in hospitalized patients? Is it possible to show these data?

The authors provide information on smoking in this group, but what about alcohol consumption (before and after COVID-19)? This could be a potentially important variable related to cognitive and emotional functioning, esp. in participants with long-COVID-19 effects.

6. PLOS authors have the option to publish the peer review history of their article (what does this mean?). If published, this will include your full peer review and any attached files.

Reviewer #1: **Yes: **Ana Sofia Costa

Reviewer #2: No

---

## [Author Response · Author response to Decision Letter 0]

23 Feb 2022

Responses to the reviewers

(please note that a file containing the reviewers comments [black text] and our responses [blue text] has been appended to this submission. We hope that this will be accessible, as we feel it will be much more 'user friendly' to read.

Reviewer #1: The authors present results from a single-center observational study of 205 consecutive patients ca. 6 months after probable to secured SARS-CoV-2 infection, with the aim of identifying deficits compatible with a post-COVID syndrome. They used a comprehensive clinical assessment with focus on cognitive, cardiac and pulmonary function, using both objective measures and PROMs. Although the characteristics of the sample may somehow limit generalization, which is correctly discussed by the authors as a possible limitation, their homogenous setting assures a comprehensive assessment with few missing data. The results are nonetheless in line with previous evidence of frequent prolonged symptomatology even in less severe cases of SARS-CoV-2 infection and in absence of significant pre-morbid conditions.

Introduction

1. The introduction would benefit from referencing more recent publications with similar aims originating from patients with confirmed infection. 

Thank you for this comment. This section has been updated and considerably expanded to both update and contextualise the reported findings.

Methods

2. Although ethical review board approval is mentioned, it is not clear if/ how participants were informed about and consented study participation, particularly given that at least one participant was assumedly underage (age rage 17-61). 

All the patients whose data are reported were full time serving members of the British Armed Forces. It is possible to join the Armed Forces in the UK within 9 months of an 18th birthday. This accounts for the fact that one patient was 17 years old. This fact is also recognised by the MOD research ethics committee – so that as long as 17-year-olds are serving, they can be included. Not to include them would be seen as a disadvantage to them in not fully representing the population.

3. Were the symptoms reported based on a priori defined listing of possible symptoms or categorized after data was collected? 

Yes. The methods have been updated to include this important fact. Thank you.

Results

4. Was there an association between the different measures and time after infection? 

We agree that this is an illuminating question, and the analysis has been expanded to address this. There was no correlation between time from acute illness to assessment for any symptom or any of the PROMS (anxiety, depression, post-traumatic stress, quality of life). There was also no correlation between this time delay and the likelihood of finding pathology on CT chest imaging or cardiac MRI scan. There was, however, a modest correlation between a small improvement in oxygen uptake on CPET and on a small improvement in FEV1 and FVC. The results section of the text has been amended to include these observations. We have also noted this in the limitations with an acknowledgement that the time-lung function and time-VO2 interactions may reflect recovery, they may also reflect a referral bias, with sicker patients being identified and referred more rapidly.

5. Table 3: It would be interesting to check if frequency of psychiatric history significantly differs of pre- vs post-COVID (current)? 

There is no difference in the frequency pre- and post-COVID. This has been included in table 3.

6. Table 3: Table legend should be revised and all abbreviations should be explained. 

Thank you for identifying this omission, it has been corrected.

7. Was there information regarding timing of symptoms? I.e., do they represent ongoing symptoms since infection or appeared after the acute phase? 

Patients were questioned at the time of their follow up about acute and ongoing symptoms. In terms of ongoing symptoms we have only collected data about whether the symptom is present or absent at follow up.

8. Table 4: It is not stated what MH stands for and it could be clarified on which guidelines the associated cut-offs (GAD-7, PHQ-9, PCL-5, Fatigue) are based on. 

Thank you for these notes. The abbreviation definition omission error in Table 4 has been rectified. Regarding referencing the thresholds of severity/for referral in different PROMS questionnaire tools, the relevant section of the methods has been extended to provide all this detail. The references have also been cited from the Table 4 legend.

9. It should be clarified that the value given for EQ5D is the visual analog scale (VAS) and no index values are presented, which would be preferable. 

This has been clarified, and referenced, in the relevant section of the methods.

10. For the results presented in Figure 3, it is not clear how cognitive dysfunction was defined (e.g., based on a priori set of deficits or associated severity, by domain/test). 

A statement clarifying that differences indicated are of statistically significant numerical group difference has been added. We have also added in the relevant section of the methods (in addition to Table 4) that a value of more than 1.5 SD below the mean is taken to indicate clear significant clinical impairment.

11. It would be interesting to check if performance status (WHO Performance status and/or FAA score) is associated with other available measures, namely cognitive dysfunction, fatigue, lung and cardiac function. 

Thank you for this suggestion. We have conducted the analysis suggested. The results have been included in the results section for performance status.

Discussion

12. Similar to the introduction, the manuscript would benefit if authors relate their results with recently published data on similar groups. 

Thank you. We have done as suggested and the discussion has been improved in this way.

 

Reviewer #2: Thus, the manuscript under review is a very important since it shows chronic health disorders associated with COVID-19 in previously healthy young or middle-aged people. The cohort consists of 205 employees of military institution, mainly men. The subjects were thoroughly tested in terms of somatic functions. A standardized set of neuropsychological tests was used to study cognitive function (NIH-TB). The results show that in a significant proportion of participants, symptoms of fatigue, cognitive and emotional disturbances persist several months after COVID-19, even if the disease was not severe and there are no significant pathological changes in organs such as the lungs or the heart.

Despite my initial enthusiasm, the manuscript raises many doubts, as below.

• The title of the manuscript and the actual content

By reading the title of manuscript, the reader is looking forward to results showing the relationship between medical findings and post-Covid-19 neurocognitive impairment. Certain medical indices are expected to be significantly related to the state of cognition. There is no such data presented. The data from the medical measurements and the results of the NIH-TB battery test are reported independently to characterize the number of participants with abnormalities. It is not known if and what medical data or other variables (fatigue, emotional disturbance) may indicate cognitive impairment after COVID-19 in this group. The key would be either to present these analyzes or to change the title. 

We take this central criticism very seriously. Additional analyses have been conducted to explore for associations between the cognitive scores and symptoms; PROMS; medical findings and exercise capacity. The findings have been summarised in the result text and are reported in an additional table (Table 5). 

• Cognitive disorders and reference measures in conclusions

There are no data and analyzes that would justify referring these cognitive disorders to the disorders detected in drunk drivers (according to the alcohol consumption limit for the UK) or the reduction of cognitive functions to the level typical for people 10 years older. Perhaps these comparisons are fully justified, but in order to draw such conclusions, the authors should refer to relevant publications or analyze the results of NIH-TB performance by drunk drivers or the standardized NIH-TB results for a group 10 years older. 

Weissenborg et al. relates the degree of deficit in the fluid composite of cognition to an administered alcohol bolus dose of 0.8 mg/kg bodyweight. This alcohol dose causes a deficit in the fluid composite akin to that seen in our patients with persistent symptoms of cognitive impairment and mood disturbance. To this citation, we have now added the references of Rajendram et al and Grant et al. These references demonstrate that a dose of 0.8 or 0.75 mg/kg mg/kg result in a blood and/or breath alcohol concentration at the UK alcohol limit for driving. 

By the way, what is the UK drink driving limit? Why the readers are supposed to know it? 

80 mg of alcohol per 100 mL of blood (the 'blood limit') 35 micrograms per 100 millilitres of breath (the 'breath limit'). This is the same as the legal limit in the USA. The mean European drink drive limit for alcohol is slightly lower at 50 mg alcohol per 100 mL blood.

• Measurement of cognitive functions and data analysis

The authors reduce the results of the NIH-TB to two components: crystallized and fluid. They define the fluid component as "ability to ... deal with complex information", the crystallized component: as learning and knowledge acquired through life. The definitions are not clear since the tasks involving crystallized functions may also involve dealing with complex information.

Moreover, the division into crystallized and fluid components is quite archaic and tells little about complex cognitive functions such as attention, executive functions, learning and memory, spatial orientation, information processing speed, and language. 

Whilst we would respectfully submit that the crystallised, fluid distinction is established, and not archaic, we do agree that a complete, and more in depth analysis, of the cognitive data (including full subdomain analysis by category of cognitive function) in this cohort will be valuable. It has always been our intention to deliver this analysis, and the work is underway. The discrete and important message of this manuscript is that there is a clear signal of impact upon the fluid component, with relative preservation of the crystallised component. This simple, clear, signal does shed light upon the nature of the long-COVID insult.

The need for a detailed analysis of the results of the cognitive function examination can be illustrated by the following example. A reduction in the efficiency of NIH-TB was found, mainly in the "fluid component" but not the "cristalised one". Language abilities depending on education, lifelong learning are usually included in the crystallized component of intelligence. However, neuropsychological studies on people after COVID-19 show that at least some aspects of language functions (naming, verbal fluency, semantic abilities) may be impaired in an effect on COVID-19. How, then, to understand the preserved vs. impaired abilities in the studied group? The results of available neuropsychological studies on COVID-19 should be presented in detail in the Introduction part and carefully discussed in the Discussion section. 

Again, we do recognise the value of a categorical analysis, which is planned. We have expanded the introduction to reflect the work already undertaken. We do also recognise that the requirements of fluent use of language do not lie completely within the crystallised component of cognition. 

• Use of abbreviations

Text and tables in many places are confusing by using abbreviations without explanation. Also, the use of abbreviation “HRCT” (high resolution CT) without specifying the organ (lungs, heart, brain) is completely incomprehensible (see Abstract). 

We agree. This has been corrected.

• Other comments

Were neurological and neuroimaging examinations included in this comprehensive medical assessment, esp. in hospitalized patients? Is it possible to show these data? 

No. We do not have brain imaging data.

The authors provide information on smoking in this group, but what about alcohol consumption (before and after COVID-19)? This could be a potentially important variable related to cognitive and emotional functioning, esp. in participants with long-COVID-19 effects. 

We agree with this. Regrettably, we do not have objective data regarding alcohol use pre-COVID. All patients completed an alcohol audit score (Table 4). This score runs from 0-40 and a total >7 is taken to indicate likely harm. The median (IQR) was 4 (2-5). 13% of participants had a score >7.

---

## [Decision Letter · Decision Letter 1]

8 Apr 2022

Comprehensive clinical assessment identifies specific neurocognitive deficits in working-age patients with long-COVID.

PONE-D-21-34299R1

Dear Dr. Holdsworth,

We’re pleased to inform you that your manuscript has been judged scientifically suitable for publication and will be formally accepted for publication once it meets all outstanding technical requirements.

Kind regards,

Lucette A Cysique, PhD

Academic Editor

PLOS ONE

Reviewers' comments:

Reviewer's Responses to Questions

**Comments to the Author**

1. If the authors have adequately addressed your comments raised in a previous round of review and you feel that this manuscript is now acceptable for publication, you may indicate that here to bypass the “Comments to the Author” section, enter your conflict of interest statement in the “Confidential to Editor” section, and submit your "Accept" recommendation.

Reviewer #1: All comments have been addressed

Reviewer #2: All comments have been addressed

2. Is the manuscript technically sound, and do the data support the conclusions?

Reviewer #1: Yes

Reviewer #2: Yes

3. Has the statistical analysis been performed appropriately and rigorously? 

Reviewer #1: Yes

Reviewer #2: Yes

4. Have the authors made all data underlying the findings in their manuscript fully available?

Reviewer #1: Yes

Reviewer #2: Yes

5. Is the manuscript presented in an intelligible fashion and written in standard English?

Reviewer #1: Yes

Reviewer #2: Yes

6. Review Comments to the Author

Reviewer #1: All comments have been address and the additional analyses do improve the manuscript by exploring possible association between the various outcomes.

Reviewer #2: I carefully read the authors' answers and the revised article. All my comments have been taken into account. I accept this version of the paper.

7. PLOS authors have the option to publish the peer review history of their article (what does this mean?). If published, this will include your full peer review and any attached files.

Reviewer #1: **Yes: **Ana Sofia Costa

Reviewer #2: No

---

## [Editor Report · Acceptance letter]

1 Jun 2022

PONE-D-21-34299R1 

Comprehensive clinical assessment identifies specific neurocognitive deficits in working-age patients with long-COVID. 

Dear Dr. Holdsworth:

I'm pleased to inform you that your manuscript has been deemed suitable for publication in PLOS ONE. Congratulations! Your manuscript is now with our production department. 

Kind regards, 

on behalf of

Dr. Lucette A Cysique 

Academic Editor

PLOS ONE